# Androgen Receptor-Mediated Transcription in Prostate Cancer

**DOI:** 10.3390/cells11050898

**Published:** 2022-03-05

**Authors:** Doğancan Özturan, Tunç Morova, Nathan A. Lack

**Affiliations:** 1School of Medicine, Koç University, Istanbul 34450, Turkey; dozturan15@ku.edu.tr; 2Koç University Research Centre for Translational Medicine (KUTTAM), Koç University, Istanbul 34450, Turkey; 3Vancouver Prostate Centre, Department of Urologic Sciences, University of British Columbia, Vancouver, BC V6H 3Z6, Canada; tmorova@prostatecentre.com

**Keywords:** androgen receptor, enhancers, prostate cancer, gene transcription, 3D genome organization, AR cistrome

## Abstract

Androgen receptor (AR)-mediated transcription is critical in almost all stages of prostate cancer (PCa) growth and differentiation. This process involves a complex interplay of coregulatory proteins, chromatin remodeling complexes, and other transcription factors that work with AR at *cis*-regulatory enhancer regions to induce the spatiotemporal transcription of target genes. This enhancer-driven mechanism is remarkably dynamic and undergoes significant alterations during PCa progression. In this review, we discuss the AR mechanism of action in PCa with a focus on how *cis*-regulatory elements modulate gene expression. We explore emerging evidence of genetic variants that can impact AR regulatory regions and alter gene transcription in PCa. Finally, we highlight several outstanding questions and discuss potential mechanisms of this critical transcription factor.

## 1. Androgen Receptor in Prostate Cancer

Prostate cancer (PCa) is one of the leading causes of cancer-related death in men [1]. In almost all PCa patients, the androgen receptor (AR) is the primary driver of growth and differentiation [2]. Given this critical role, AR pathway inhibitors (ARPI) are the standard of care for treating patients with recurrent or metastatic forms of the disease [3,4]. However, while treatment is initially successful, ~20% of patients develop resistance and progress to a castration-resistant prostate cancer (CRPC) [5]. This aggressive form of the disease is invariably lethal. Intriguingly, the AR still remains active in the majority of resistant patients through various mechanisms, including AR point mutations [6], constitutively active AR variants [7], and most commonly, AR gene and enhancer amplification [8,9,10,11]. Despite the importance of AR-mediated transcription in PCa, fundamental aspects of how this nuclear receptor drives gene expression are only now being revealed. In this review, we summarize the mechanism of AR-mediated transcription in PCa and discuss outstanding questions.

## 2. Androgen Receptor-Mediated Gene Transcription

The AR is a 919 amino acids (110 kDa) protein that contains an N-terminal domain (NTD), DNA-binding domain (DBD), and C-terminal ligand-binding domain (LBD) [12,13]. The inactive apo-form of AR primarily resides in the cytoplasm, where it is stabilized by chaperone proteins. When activated by androgens, most commonly testosterone or the more potent metabolite 5α-dihydrotestosterone (DHT), the AR undergoes an allosteric modification, homodimerizes, and then translocates into the nucleus where it binds to DNA at AR binding sites (ARBS) [14]. The location of these ARBS is influenced by numerous features including the DNA primary sequence or motif, protein–protein interactions, transcription factor (TF) occupancy, and chromatin accessibility. Once bound to enhancer *cis*-regulatory elements (CREs), the AR recruits coregulators [15], remodeling complexes [16], and other TFs [17,18,19,20] to create a transcriptional hub that initiates an AR-dependent transcriptional program, which impacts the expression of several hundred target genes [21]. These genes contribute to proliferation, cellular differentiation, and potentially metastasis. The specific genes associated with these complex cellular processes are controversial, though several have been proposed, including *c-Myc*, *EVT1*, and *EIF5A2* (eukaryotic translation initiation factor) [22,23,24]. When looking at essentiality from published genome-wide CRISPR screens of AR-regulated genes in PCa cells (LNCaP) (Figure 1), we find many known and novel essential genes, including coactivators such as *GRHL2* (grainyhead-like transcription factor 2) [25]; metabolic genes such as *DNM1L* (dynamin-related protein 1) [26], SREBP (sterol regulatory element-binding protein) cleavage activating protein *SCAP* [27] and *mTOR* (the protein kinase mammalian target of rapamycin) [28]; and transcriptional regulators such as *NFKBIA* (NFKB inhibitor alpha), an inhibitor protein of NF-κB and p53 [29].

## 3. Pioneer Factors and DNA Binding 

Before the AR binds to DNA, the ARBS is first “primed” by pioneer factor (PF) proteins that interact with heterochromatin and increase chromatin accessibility [30]. Given that PFs determine where the AR can potentially bind, these proteins strongly influence ARBS locations [31,32]. PFs engage with nucleosomes [33] that are dynamically transitioning between fully wrapped and transient exposure states [34,35]. The window of exposure is sufficient to allow binding of PFs as well as other TFs [36]. PFs typically bind to poised/active enhancers containing histone marks such as histone 3 lysine 4 mono/di-methylation (H3K4me1/me2) [37,38] and histone 3 lysine 9/27 acetylation (H3K9/27ac) [39,40]. There is limited PF binding at regions with strong silencing/repressive histone marks and DNA methylation [41,42]. While not reported with AR, other steroid receptors have been shown to recruit ATP-dependent modifiers that can interact with closed chromatin independent of PFs [43,44] and recruit these to target sites [45]. PFs are classified based on their protein domains and mechanism of action [45,46,47]. FOXA1 (forkhead box transcription factor a1) plays a critical role in the activity of AR and other nuclear receptors [48]. FOX proteins contain a winged helix DBD domain that allows recognition of full/partial DNA motifs in the nucleosome [49,50,51]. The C-terminal of FOXA1 is necessary for both unwrapping the chromatin through an ATP-independent mechanism and recruiting ATP-dependent chromatin modifiers [52]. In AR-mediated transcription, FOXA1 both opens chromatin for direct AR binding and also acts as an anchor for AR to indirectly bind CREs [53]. Demonstrating its critical role, knockdown of FOXA1 causes a decrease in overall AR binding with a massive redistribution of ARBS at tens of thousands of new sites [32]. This is strongly influenced by AR itself, as those sites that are unaffected following FOXA1 knockdown generally have higher AR occupancy [54]. GATA2 (GATA-binding factor 2) is another well-characterized PF that increases accessibility at ARBS [36,55]. Although GATA2 chromatin accessibility induction is less effective than FOXA1 [52], GATA proteins facilitate binding of AR and estrogen receptor (ER) in prostate [56] and breast cancer [57], respectively. Different from FOXA1, the GATA family of TFs requires other chromatin remodeling proteins, such as the SWI/SNF complex, to alter accessibility [45,58,59]. Lastly, HOXB13 (Homeobox b13), a member of the HOX family of proteins, has been speculated to have potential PF activity due to its preference to bind methylated DNA that is found at heterochromatin [60,61]. However, further work is needed to demonstrate this potential function. While the hierarchy of PFs is not clearly defined during AR activation, >70% of ARBS overlap with either GATA2, FOXA1, or HOXB13 binding sites [53,56,59,62]. Given that PF activity is critical for the oncogenic transformation of several cancers, including prostate and breast, there is ongoing research to develop therapeutics that target these PFs [63,64].

## 4. Plasticity of the AR Cistrome in Prostate Cancer Progression

AR induces gene transcription by binding to specific CREs in the genome. Identifying the location of ARBS is therefore critical to understand how AR functions. The first ARBS identified were found at the promoter of the rat probasin [65,66] and *KLK3/PSA* (kallikrein related peptidase 3/prostate specific antigen) gene [67,68,69]. The identification of subsequent ARBS, most prominently the AREIII (AR regulated enhancer III) upstream of *KLK3/PSA*, demonstrated that AR primarily drives transcription through enhancers rather than promoters [70,71]. Enhancers are non-coding regulatory elements that are required for gene expression, as they “enhance” transcription of target genes [72,73]. AR-driven enhancer activity has been confirmed with various genes including *PSMA* (prostate-specific membrane antigen) [74] and *p21* [75]. Large-scale functional genomic studies, first with ChIP-on-chip (chromatin immunoprecipitation followed by microarray) and then ChIP-seq (chromatin immunoprecipitation followed by sequencing), provided additional support that AR activity occurred through enhancer CREs as these studies demonstrated that the vast majority of ARBS (>95%) are located in non-coding intronic or intergenic regions, with few binding sites found at promoters (<2%) [76,77,78].

The AR cistrome, or genome-wide binding sites, is not static, and clinical ARBS display remarkable plasticity and significant reprogramming during both tumor initiation and disease progression [79]. During neoplastic development, there is a dramatic expansion (3×) of ARBS in primary PCa compared to normal prostate [53]. Similarly, in metastatic CRPC samples, the AR gains an additional > 17,000 distinct binding sites that are associated with prostate development [79]. This suggests that CRPC regains an AR-driven early developmental transcription signature to potentially increase survival and proliferation during ARPI treatment. Interestingly, the changes in the AR cistrome seem unlikely to be solely due to chromatin accessibility as in both normal prostate and primary PCa most gained sites are already accessible euchromatin bound by PFs [79]. The reason why AR does not bind to these accessible regions in primary PCa remains an active area of research. While speculative, the increased expression of AR or coregulators in advanced stages of the PCa may influence where and for how long AR binds to chromatin. Supporting this potential mechanism, overexpression of *AR in vitro* has been shown to sensitize binding and alter ARBS [80]. Further, changes in *FOXA1* expression alter the global chromatin accessibility and generate both pseudo-AR hypersensitivity and an increase in open chromatin ARBS [54]. This correlates with the broad increase in chromatin accessibility that is observed during the progression from primary PCa to CRPC [81,82]. Potentially, these newly accessible regions primed by FOXA1 or other coregulatory proteins may require increased expression of AR to compensate for the gained regions.

## 5. Impact of Motifs on AR Binding and Activity

ARBS are enriched for a conserved androgen response element (ARE) binding motif that is made up of two 6-bp asymmetrical elements separated by a 3-bp spacer (5′-AGAACAnnnTGTTCT-3′) [12,69,83]. This specific motif has been extensively validated by various studies, including an *in vitro* SELEX-seq (systematic evolution of ligands by exponential enrichment followed by sequencing) with recombinant AR-DBD protein [84]. Nonetheless, it remains unclear how important an ARE motif is to AR binding *in situ* [85]. Whole-genome AR ChIP-seq shows that only 8–30% of ARBS contain a canonical ARE, while most binding sites have a more common ARE half-motif (5′ -AGAACA-3′) [53,56,78,79,86]. However, the importance of ARE half-motifs is itself subject to discussion. A previous study found no evidence for binding at half-motif sites [87], whereas other work has proposed two modes of AR binding that include both half- and full-motif stabilized by FOXA1 interactions [30,88]. Further, there is contradicting data on the impact of motifs on enhancer activity. Studies with glucocorticoid receptor (GR) observed higher enhancer activity and target gene expression at those regions with GR motifs [89]. However, this was not binary, and many enhancers harboring weak GR motifs also had similar enhancer activity as those with strong motifs [89]. Our recent work did not report a strong correlation between any specific ARE motifs and AR enhancer activity [90]. These findings suggest that there is mixed evidence supporting the prerequisite of an ARE motif for AR binding and enhancer activity. Incorporating datasets such as chromosome accessibility (ATAC-seq, DNA-seq, FAIRE-seq) or alternative methodologies such as CUT&RUN may potentially help to reduce noise in defining *in situ* AR motifs.

## 6. Impact of Chromatin Modifying Enzymes on AR Activity

Once bound to DNA, the AR recruits chromatin-modifying enzymes that stabilize accessibility and provide a platform for coregulators and TFs [91]. This includes histone methyltransferases such as EZH2 [92,93,94], SET9 [95], and MLL complex (MLL, MLL4, WDR5, ASH2L) [96]; histone acetyltransferases (HAT) such as p160, SRC-1, TIF2/GRIP1-1, ACTR/AIB1/RAC3/pCIP [97,98,99,100,101], CBP [102], p300 [103], and pCAF [104]; and histone deacetylases (HDAC) such as HDAC1-3 (class I), HDAC4-10 (class II), and SIRT1-7 (class III) and HDAC11 (class IV) [105]. Characterization of the AREIII enhancer demonstrated the sequential recruitment of p160 and p300, was then followed by CBP and pCAF [21]. HAT and HDAC work antagonistically to acetylate the lysines on histone N-terminal tails that promote the formation of heterochromatin through electrostatic interactions [106]. Further, BRM (SMARCA2), a member of the SWI/SNF chromatin remodeling complex, has been shown to be essential for *KLK3/PSA* and rat probasin expression [107]. Overexpression of the related BRG1 (SMARCA4) in *BRM/BRG1* mutant cell lines showed limited AR activity at *KLK3/PSA* and rat probasin enhancers [108]. In ER-mediated transcription, FOXA1 has also been shown to recruit MLL3 to promote H3K4me3 at ER binding sites in breast cancer cell lines [109]. The kinetic hierarchy of regulation still needs further characterization given the intricacies of this dynamic transcriptional complex.

## 7. AR-Coregulators and Gene Transcription

Once bound to chromatin, AR forms a complex with numerous coregulatory proteins to activate enhancers and alter the expression of target genes (Figure 2) [89,110,111,112]. This process involves many different proteins, including coactivators such as CBP/p300 (CREB binding protein) and SRC-1 [21]; chromatin modifying enzymes that alter accessibility [108]; and proteins that stabilize AR binding [16,113]. Overall, this is an extremely dynamic process, and to date >250 proteins have been identified that interact with the AR and have potential coregulatory activity [114]. Further, AR interacts with MED1 (mediator complex subunit 1) and other members of the Mediator complex to stabilize AR-mediated enhancer–promoter interactions [17]. Recent work has suggested that these AR–MED1 interactions may induce the formation of phase condensates at super-enhancer regulatory elements [115]. These super-enhancers have higher transcriptional output of target genes than individual enhancers and play a pivotal role in cell identity and tumor progression [116]. In CRPC, several gained super-enhancers were proposed to activate oncogenes such as *CHPT1* (choline phosphotransferase 1) and drive resistance [117]. In addition to these coregulatory proteins, long non-coding RNAs (lncRNAs) and enhancer RNAs have also been reported to impact AR-mediated gene transcription [8,118]. Although this is a current subject of research, these RNA species are proposed to recruit protein complexes to the transcription target sites [119]. For example, the PCA3 (prostate cancer antigen 3) lncRNA interacts with AR and stabilizes androgen-induced transcription [120]. However, there is considerable diversity in the mechanism of action. AR-upregulated ARLNC1 (androgen receptor regulated long noncoding RNA 1) has been shown to stabilize AR mRNA transcripts and alter expression through a transcriptional feedback loop [121]. Further, SChLAP1 (second chromosome locus-associated with prostate-1) is highly expressed in CRPC and induces proliferation through interactions with the SWI/SNF chromatin-modifying complex [122].

While poorly understood, AR-mediated gene downregulation has been proposed to occur through both direct repression and indirect coactivator sequestering, also known as squelching [123]. In direct repression, corepressor proteins such as homologous proteins SMRT and NCoR (nuclear receptor corepressor) interact with AR and recruit histone deacetylases such as HDAC4 that cause chromatin compaction and transcriptional repression (Figure 3) [124]. Specific corepressors include RIP140 (receptor-interacting protein 140), which directly binds to C-terminus of AR protein [125]. Others such as LCoR (ligand dependent corepressor), inhibit AR-mediated transcription by interacting with HDACs and CtBP (C-terminal binding protein), which suppress tumor growth *in vivo* [126,127]. Calcium-binding protein calreticulin inhibits AR activity through its DBD [128,129]. In contrast with these mechanisms, the squelching model of repression proposes that the activated AR, which previously resided in the cytoplasm, binds to numerous coregulatory proteins that impact the activity of non-AR TFs by limiting access to these critical proteins. AR and other nuclear receptors are also the subject of auto-squelching, which can repress target genes [130]. 

Regardless of the mechanism, gene downregulation is believed to be important for the growth and progression of advanced stage PCa [131]. Importantly, the activation of AR leads to the downregulation of *c-Myc* [132], which has an antagonistic transcriptional network with AR [133]. *c-Myc* repression by AR is largely independent of AR binding to its target sites and primarily occurs via the redistribution of AR coactivators [134]. Further, *c-Myc* regulation by histone methyltransferase, DOT1L (disruptor of telomeric silencing 1-like), and AR through an enhancer has also been reported [135]. Decreased AR expression upon the inhibition of DOT1L, which coregulates AR and MYC pathways, leads to increased expression of AR-target genes by other TFs such as c-Myc. AR has also been shown to alter the expression of other known tumor-suppressors such as *p53*, *PTEN* (phosphatase and tensin homolog deleted on chromosome 10), and *LRIG* (leucine-rich repeats and immunoglobulin-like domains). AR inhibits *p53* expression, while p53 directly represses the expression of AR by binding to target promoters [136,137]. PI3K (phosphoinositide3-kinase) signaling is altered in PCa through loss of PTEN and is associated with aggressive PCa prognosis [138]. Expression of *PTEN* is inversely correlated with *AR* in PCa tumors, and AR is reported to directly inhibit *PTEN* expression [139]. Finally, elevated expression of the AR-stimulated tumor-suppressor *LRIG* is associated with increased overall survival in PCa cohorts [140,141]. *LRIG* expression is also affected by SUMOylation of AR in which small ubiquitin-related modifiers (SUMOs) covalently bind to the AR and alter the downstream transcription events [142]. Interestingly, motif analysis of corepressor-bound AR and coactivator-bound AR showed a similar binding motif, which suggests there may be competition between these two complexes for gene activation/repression [143].

## 8. AR Enhancers in Gene Transcription

AR primarily drives gene expression through enhancer CREs. Located at euchromatin [144], enhancers commonly correlate with histone marks such as H3K27Ac and H3K4me1 [145]. Enhancers are also typically bound by multiple TFs [89,90,146,147,148], RNA polymerase II [149], transcriptional coactivators [150,151,152], and CEBP/p300 [153]. Nonetheless, these features only broadly correlate with activity, and there are numerous enhancer CREs that do not contain some or any of these specific modifications [90,154,155,156]. Functional annotation is needed to understand how ARBS work together to drive gene transcription. This is particularly important in AR-mediated transcription, as there are 10–100× more ARBS (tens of thousands) than differentially expressed genes (hundreds). The development of novel high-throughput enhancer assays such as STARR-seq (self-transcribing active regulatory region sequencing) have enabled researchers to test the enhancer activities of thousands of genomic regions in a single experiment [157,158,159,160]. Recently, all high-confidence clinical ARBS were tested using STARR-seq and revealed three different classes of binding sites—named as inducible, inactive, and constitutive enhancers. Only a fraction of the regions showed AR-activated or inducible enhancer activity (7%), and instead the majority of ARBS did not demonstrate any enhancer activity (81%). Further, approximately 12% of ARBS exhibited constitutive enhancer activity that was independent of AR binding. Inducible AR enhancers were found to correlate with both high-AR occupancy and an increase in chromatin loops to other CREs and gene promoters. While it could be argued that these differences in activity are either contextually or temporally dependent, a strong correlation was observed between these *in vitro* annotations and H3K27Ac in clinical PCa samples. Unexpectedly, when these ARBS classes were functionally tested, both inactive and constitutively active enhancers, in addition to inducible enhancers, were frequently required for AR-mediated gene transcription. Supporting this functional role, each ARBS class demonstrated equivalent evolutionary conservation, suggesting that each enhancer type is required for gene transcription [90]. Apart from AR inducible enhancers, different mechanisms have been proposed for constitutive and inactive ARBS enhancers. Inactive sites could support long-range chromatin interactions or increase the local AR concentration to trigger gene transcription. A similar stabilization may occur with constitutive enhancers where multiple TFs support looping following AR binding. In this model, these genomic regions would produce enhancer reporter signals in an episomal assay with non-AR TF binding but would only contribute to gene transcription when the AR is bound. Large-scale functional testing of these distinct ARBS enhancer classes is needed to interrogate their role in AR-mediated gene transcription.

There is increasing evidence that multiple ARBS work together to drive gene transcription. More than 60% of the AR-regulated genes have >2 ARBS within 200 kb proximity (Figure 4a). For example, TMPRSS2 (transmembrane serine protease 2) is regulated by several enhancers bound by AR/FOXA1/p300 that loop to the gene promoter and induce transcription [161]. How this occurs is poorly understood, but several distinct biological models have been proposed to explain the collaborative mechanism between multiple bound TFs [162,163]. Recent studies of ER binding sites have shown both hierarchical and synergistic interactions between enhancers [164]. The hierarchical model suggests that a dominant motif-containing enhancer can activate gene expression by itself, and a nearby weak motif-containing enhancer only contributes secondarily to gene activation. In contrast, the synergistic model proposes that a motif-containing binding site only contributes to gene expression if a neighboring binding site is also bound. However, it remains to be determined whether a single model can explain all interactions between different ARBS enhancers.

## 9. 3D Genome Organization

AR-regulated enhancers are brought in close physical proximity to the target gene promoter by chromatin looping [165]. These ARBS enhancer–promoter loops occur within topologically associated domains (TADs) that are formed by both insulator protein CTCF (CCCTC-binding factor) and cohesin [166]. TADs segment the genome into regions that contain high contact frequency loops and similar histone modification patterns [167,168]. Within each TAD, most ARBS loops are distributed between 10 kb to 1 mb (Figure 4b), though there are notable exceptions, including the gene STEAP4 (six-transmembrane epithelial antigen of prostate 4) that interacts with an ARBS found >2 MB from the target promoter [169]. Enhancer–promoter loops in PCa cell lines are enriched for numerous binding motifs, including AR, FOXA1, and the coregulator GRHL2 [170]. These enhancer–promoter interactions are proposed to either be pre-existing or formed *de novo* [171,172]. Pre-existing links are convenient for rapid transcriptional activation [173], whereas *de novo* loops can be formed through TF interacting structural proteins such as YY1 (Yin Yang 1) [174,175]. Gene expression and loop strength are independent of the distance between an enhancer and its target promoter [176]. Work from the ENCODE project demonstrated that the average distance of enhancer–promoter loops is around 120 kb with almost four enhancers for any given active gene [177]. However, these regulatory networks are complicated by the significant alterations found in PCa at almost all hierarchical levels of chromosomal organization [178]. A recent study conducted HiC, a whole genome chromosome conformation capture assay, in multiple PCa cell lines, including RWPE-1, LNCaP, DU145, 22Rv1, VCaP, PC3, MDAPCa2a, MDAPCa2b, and C4-2B, identified 387 TAD gene compartments that were distinct for each cell line [179]. Similarly, *in situ* HiC maps of RWPE1, C4-2B, and 22Rv1 cell lines showed that common TADs found in all cell lines were much smaller than those TADs unique to a single cell line [170]. Using “normal” prostate epithelial cells and models of PCa (LNCaP and PC3), those so-called “normal” TADs were much larger, higher in number, and located in distinct positions [180]. However, low-input HiC from both primary PCa (*n* = 12) and benign prostate tissues (*n* = 5) demonstrated that, unlike cell lines, there was no significant difference in the number of TADs called or in TAD borders between samples [181]. Combining these data with whole-genome sequencing, they found only one structural variant (out of 260) with altered gene expression in an intra-TAD region. While spatial organization of the genome affects gene transcription, it remains to be determined how these changes in chromatin looping affect AR-mediated gene expression.

## 10. Enhancer CRE Mutations

PCa has a relatively low somatic mutation frequency compared with other cancer types [184]. Common oncogenic drivers include *TP53* (tumor protein 53) and *PTEN* (phosphatase and tensin homolog), as well as prostate-specific recurring mutations such as *SPOP* (speckle type BTB/POZ protein) and *FOXA1* [185]. Given the critical role of AR signaling in CRPC, late-stage PCa commonly harbors AR somatic mutations including gene duplications, single nucleotide variants (SNV), or structural variants (SV) [10]. Several excellent reviews have discussed protein coding mutations in PCa and their role to stratify patients for treatment [186,187,188]. However, protein coding regions make up only ~1% of the whole genome [189,190,191], and there is increasing evidence that non-coding mutations at enhancer CREs contribute to PCa progression. This is particularly important as while the mutational burden is relatively low in PCa, there is a high frequency of SVs that can cause enhancer-driven dysregulation of transcriptional networks [192]. For instance, duplication of an upstream enhancer that regulates the AR gene is commonly found in most advanced PCa patients (81%) and can act as the sole driver of ARPI-resistance in CRPC [193,194]. Further, there are several common fusion events where AR-driven regulatory elements induce transcription of oncogenic driver genes through enhancer hijacking [195,196,197]. This was first reported with TMPRSS2-ERG fusions [198] and then later TMPRSS2-ETV1/ETV4 fusions [199]. *TMPRSS2* is a prostate-specific AR-regulated gene, whereas *ERG* is a critical regulator of proliferation, differentiation, and apoptosis [198,200]. As a result of the fusion, the *ERG* gene expression becomes regulated by AR signaling and is highly expressed in PCa. Similar complex rearrangements between the AR-regulated gene *NRF1* and *BRAF* have also been observed [201]. *n-Myc* and *c-Myc* expression have also been attributed to enhancer hijacking of distal enhancers in neuroblastoma [202,203]. Changes in gene expression can also occur by mutations that alter TAD structures. A commonly found deletion in the 17p13.1 locus that contains the tumor-suppressor *p53* gene separates a well-defined TAD that occurs in normal cells into two distinct TADs, with significant changes in CRE usage [180]. Further, a recent study demonstrated that disruption of a single CTCF binding site in the KLK locus alters transcription of the gene cluster [204]. However, it is not known exactly which enhancer regions play a role in this activation. Further, this potentially may be locus-specific, as there is evidence that CTCF depletion is not affecting enhancer–promoter connections [205]. Large-scale chromosomal alterations can also cause circular extrachromosomal DNA (ecDNA) that leads to the expression of numerous oncogenes through changes in enhancer usage [206,207]. Given the highly unstable genomic landscape of PCa, ecDNA is increasingly being found [208,209]. Further, enhancer retargeting caused by promoter somatic mutations can also lead to gene reactivation [210].

There is conflicting evidence for the role of non-coding SNVs in PCa initiation and progression. The vast majority of PCa point mutations are non-coding and found in intergenic (46%), intronic (44%), and promoter (9%) regions [211]. Most of these SNVs are likely passenger mutations. In the large-scale Pan-Cancer Analysis of Whole Genomes (PCAWG), only 0.3% (986 of 276,892) of patients had recurrent non-coding mutations, suggesting that there is little selective pressure [211]. However, this interpretation is complicated by the nature of gene transcription, where multiple CREs commonly work together and a mutation at independent enhancer regions could potentially cause the same alteration in gene expression [212]. Therefore, instead of recurrent individual mutations causing transcriptional dysregulation, they could occur at multiple sites in a local regulatory network (plexuses) and alter the expression of critical genes [211]. Supporting a potential role of CRE SNVs in PCa development, the majority of single nucleotide polymorphisms (SNPs) identified from PCa-associated genome-wide association studies (GWAS) occur in non-coding regulatory regions and are proposed to alter TF binding at enhancer regions [213,214,215]. While the impact of these mutations is unclear, we and others have demonstrated that the binding sites of lineage-specific TFs have an increased rate of somatic mutations [211,216]. However, interpretation of these SNVs is limited by poor understanding of enhancer “grammar” that prevents the identification of potential pathogenic variants.

## 11. Targeting AR and Coregulators

Current PCa therapies target AR through either direct antagonism (bicalutamide, enzalutamide, apalutamide, etc.) or by reducing the synthesis of androgenic steroids (LHRH agonists, abiraterone, etc.) [217]. However, while initially effective, almost all tumors eventually develop resistance to treatment [5]. While a subset of these resistant tumors differentiates into a neuroendocrine state (<15%), the vast majority of CRPC tumors still remain dependent on AR signaling. Given their critical function, AR-coactivator interactions have been proposed as an alternative pharmacological target for overcoming many common mechanisms of resistance [218].

## 12. Conclusions

In this brief review, we discuss the mechanism of AR-regulated gene expression in PCa. Numerous studies characterizing the AR cistrome have revealed that the vast majority of ARBS are located at enhancer CREs that regulate the transcriptional activity via chromatin looping. However, while these ARBS are well characterized, there are still many outstanding questions, particularly, related to the expansion of ARBS in CRPC where there is a broad reactivation of early developmental transcriptional processes. There is emerging evidence that ARBS can influence gene transcription even without episomal enhancer activity, suggesting that AR directly or via other TFs can potentially stabilize CRE chromatin interactions and influence transcription. From this perspective, there is a need for additional chromosomal genome organization datasets to improve our understanding of phenotypic events in the different stages of PCa. With high-throughput dataset initiatives such as ENCODE [191,219] and 4D-Nucleome Project [220], as well as several large-scale clinical projects, these datasets will help to contribute to our knowledge of this complex process. By understanding AR-mediated gene transcription, we can both begin to stratify potential non-coding driver mutations and identify therapeutic vulnerabilities to better treat late-stage PCa patients.

## Figures and Tables

**Figure 1 cells-11-00898-f001:**
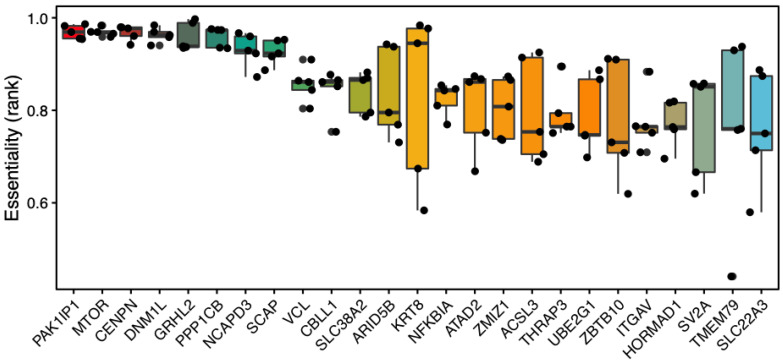
Essential and AR-upregulated genes in androgen-sensitive and PSA-positive LNCaP cell line. Data taken from 5 LNCaP genome-wide CRISPR screens in DepMap database (DepMap 21Q4, DepMap 21Q3, GeCKO 19Q1, GeCKO CERES, Sanger CERES) and ranked based on their essentiality. AR-upregulated genes are taken from RNA-seq samples of androgen-treated LNCaP cells (GEO: GSE64529).

**Figure 2 cells-11-00898-f002:**
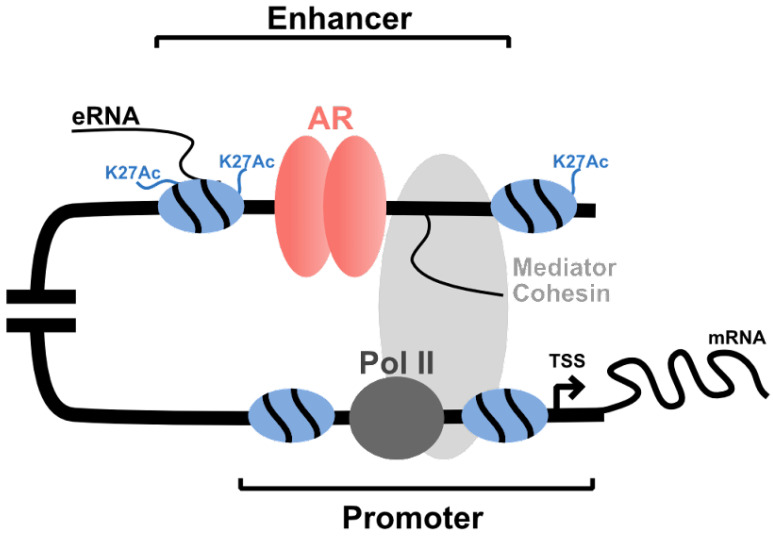
Cartoon representation of ARBS enhancer activity on AR-mediated gene. Upon AR binding, coactivators, mediator complex, cohesin proteins, and transcriptional machinery are recruited to initiate gene expression.

**Figure 3 cells-11-00898-f003:**
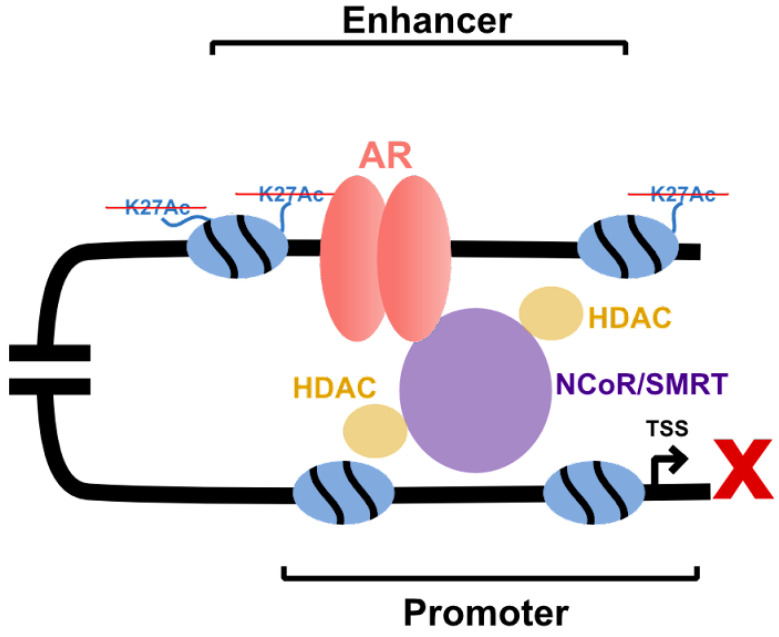
AR-mediated gene repression. AR is bound by corepressors such as NCoR/SMRT, creating a corepressor complex and facilitating HDAC activity to suppress gene activation.

**Figure 4 cells-11-00898-f004:**
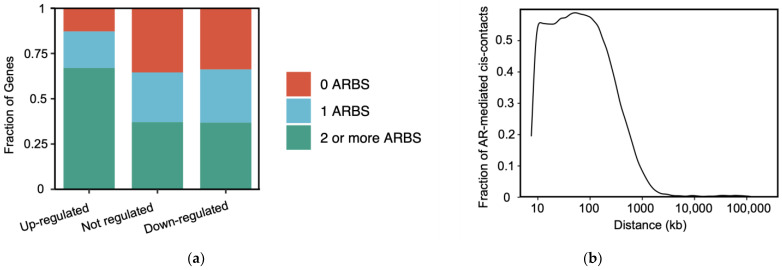
ARBS interaction landscape. (**a**) Multiple ARBS reside in close proximity to AR-regulated genes. Frequency of ARBS were quantified using publicly available androgen-induced RNA-seq [182] and AR ChIP-seq [183] from LNCaP cells. (**b**) AR ChIA-PET *cis*-contacts are mostly concentrated around 10 kb to 1 mb. AR-mediated looping dataset were used calculate the frequency of chromatin loops in VCaP cells [169].

## Data Availability

Not applicable.

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
