# Peer review of "Androgen Receptor-Mediated Transcription in Prostate Cancer"

_cells, 2022, doi:10.3390/cells11050898_

Round 1

Reviewer 1 Report

  1. Inthis review, author discussed in detailthe mechanism of AR-mediating transcription. of downstream genes in prostate cancer. Given that a few similar articles have been published recently, this review lack enough innovation.
  2. The Figure 1, showing a biological process, needs to give a better explanation in the. text.
  3. There are several formatting errors, like“8.3. D genome organization” in line 230.There should be spaces in line 231.
  4. Author describes how AR regulates downstream transcription, without mentioning how this mechanism affects the outcome of prostate cancer, like proliferation, differentiation or metastasis.
  5. In line 161-162, long non-coding RNAs (lncRNAs) have been reported to impact AR-mediated gene transcription. The author should give one or two examples to explain how lncRNAsregulates AR-mediatedtranscription.
  6. In line 162, although it is notfully understood, which does it mean?
  7. In line 192: “our group” needs to be clarified.
  8. Currently, which therapies are available to treat PCa against the mechanism of AR-mediated gene transcription? Clinical implication and challenges also need to be discussed.

Reviewer 2 Report

The review by Özturan et al. summarizes the transcriptional activity of the androgen receptor. Authors include pioneering factors, changes of the cistrome during prostate cancer progression, AR binding site motifs including enhancers and 3D nuclear genome changes. This topic is interesting since many novelties in this field were identified in recent years.

The review is superficial and must be enhanced in several aspects.

Important aspects of the genomic activity of the AR are lacking.

Major points:

  1. An important aspect of AR mediated transcription by chromatin modifiers is missing. The interaction and recruitment of chromatin modifying enzymes to AR is missing. An additional figure of AR interacting coregulators that harbour histone modifying enzymatic activity would be very useful for the general reader.

  1. Figure 1 depicts very simplified only the AR-mediated transcriptional activation. However, a great proportion of genes regulated by AR are repressed by androgens. Although various mechanisms are known, this important aspect must be more emphasized. Therefore, another figure that addresses gene repression by AR must be added.

  1. Also, besides NCoR and HDACs other AR corepressors are known that mediate repression by AR. This chapter must be enlarged as well. Similarly, mention interaction with key coactivators.

  1. Indicate the molecular mechanisms how pioneering factors allow AR recruitment to chromatin.

  1. Explain in more detail the potential contradiction in the text on page 2:

“the knockdown of FOXA1 causes a decrease of overall AR binding with a massive redistribution of ARBS and tens of thousands of new sites. Further, loss of FOXA1 increases the binding affinity of AR at existing ARBS”. Mention more precisely what is meant by existing. The reduced ARBS number due to knowckdown?

  1. Authors mention c-MYC as an important gene regulated by AR. Please add and discuss novel contributions in this field: e.g. Guo et al., Nat Commun 2021; Vatapalli et al., Nat Commun, 2020.

  1. Authors mention interaction of AR with c-Myc as an oncogene. Add a chapter of interaction of AR with tumor suppressors such as LRIG, p53 and others.

Minor point:

Remove the dot in the title on page 5:  “8.3. D genome organization“

Round 2

Reviewer 1 Report

I agree with the publication of the paper.

Reviewer 2 Report

Authors addresssed in full satisfactory manner the critical points.

Minor point:

There seems to be one typo in line 52: It should be ETV1 instead of EVT1.